# Gaussian Graph Network: Learning Efficient and Generalizable Gaussian Representations from Multi-view Images

**Shengjun Zhang, Xin Fei, Fangfu Liu, Haixu Song, Yueqi Duan**[*]
Tsinghua University
{zhangsj23, feix21}@mails.tsinghua.edu.cn, duanyueqi@tsinghua.edu.cn

## Abstract

3D Gaussian Splatting (3DGS) has demonstrated impressive novel view synthesis performance. While conventional methods require per-scene optimization, more recently several feed-forward methods have been proposed to generate pixel-aligned Gaussian representations with a learnable network, which are generalizable to different scenes. However, these methods simply combine pixel-aligned Gaussians from multiple views as scene representations, thereby leading to artifacts and extra memory cost without fully capturing the relations of Gaussians from different images. In this paper, we propose Gaussian Graph Network (GGN) to generate efficient and generalizable Gaussian representations. Specifically, we construct Gaussian Graphs to model the relations of Gaussian groups from different views. To support message passing at Gaussian level, we reformulate the basic graph operations over Gaussian representations, enabling each Gaussian to benefit from its connected Gaussian groups with Gaussian feature fusion. Furthermore, we design a Gaussian pooling layer to aggregate various Gaussian groups for efficient representations. We conduct experiments on the large-scale RealEstate10K and ACID datasets to demonstrate the efficiency and generalization of our method. Compared to the state-of-the-art methods, our model uses fewer Gaussians and achieves better image quality with higher rendering speed.

## 1 Introduction

Novel view synthesis is a fundamental problem in computer vision due to its widespread applications, such as virtual reality, augmented reality, robotics and so on. Remarkable progress has been made using neural implicit representations [30, 39, 40], but these methods suffer from expensive time consumption in training and rendering [26, 12, 1, 37, 60, 10, 15, 31]. Recently, 3D Gaussian Splatting (3DGS) [19] has drawn increasing attention for explicit Gaussian representations and real-time rendering performance. Benefiting from rasterization-based rendering, 3DGS avoids dense points querying in scene space, so that it can maintain high efficiency and quality.

Since 3DGS relies on per-subject [19] or per-frame [28] parameter optimization, several generalizable methods [63, 6, 4, 41] are proposed to directly regress Gaussian parameters with feed-forward networks. Typically, these methods [6, 4] generate pixel-aligned Gaussians with U-Net architectures, epipolar transformers or cost volume representations for depth estimation and parameter predictions, and directly combine Gaussian groups obtained from different views as scene representations. However, such combination of Gaussians leads to superfluous representations, where the overlapped regions are covered by similar Gaussians predicted separately from multiple images. While a simple solution is to delete redundant Gaussians, it ignores the connection among Gaussian groups. As

---

[*]Corresponding author.

38th Conference on Neural Information Processing Systems (NeurIPS 2024).

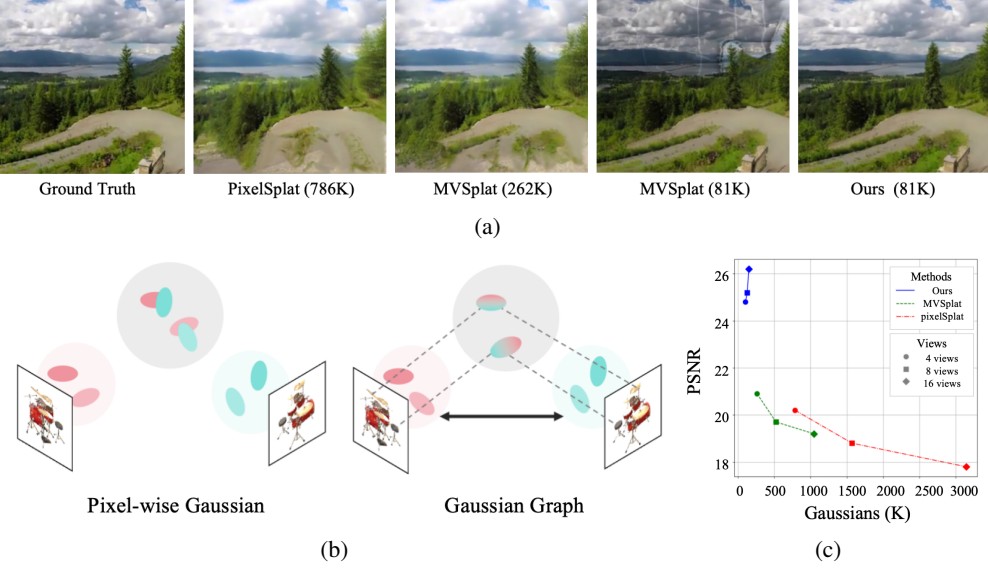

Figure 1: Comparison of previous methods and ours. (a) We visualize the rendering results of various methods and report the number of Gaussians in parentheses. (b) Previous pixel-wise methods can be considered as a degraded case of Gaussian Graphs without edges. (c) We report PSNR as well as the number of Gaussians for pixelSplat [4], MVSplat [6] and GNN under different input settings.

illustrated in Figure 1a, pixelSplat [4] and MVSplat [6] suffer from artifacts with several times as many Gaussians as ours, while the deletion of similar Gaussians hurts rendering quality.

To tackle the challenge, we propose Gaussian Graphs to model the relations of Gaussian groups from multiple views. Based on this structure, we present Gaussian Graph Network (GGN), extending conventional graph operations to Gaussian domain, so that Gaussians from different views are not independent but can learn from their neighbor groups. Precisely, we reformulate the scalar weight of an edge to a weight matrix to depict the interactions between two Gaussian groups, and introduce a Gaussian pooling strategy to aggregate Gaussians. Under this definition, previous methods [4, 6] can be considered as a degraded case of Gaussian Graphs without edges. As shown in Figure 1b, our GGN allows message passing and aggregation across Gaussians for efficiency representations.

We conduct extensive experiments on both indoor and outdoor datasets, including RealEstate10K [64] and ACID [24]. While the performance of previous methods declines as the number of input views increases, our method can benefit from more input views. As shown in Figure 1c, our model outperforms previous methods under different input settings with higher rendering quality and fewer Gaussian representations. Our main contributions can be summarized as follows:

• We propose Gaussian Graphs to construct the relations of different Gaussian groups, where each node is a set of pixel-aligned Gaussians from an input view.

• We introduce Gaussian Graph Network to process Gaussian Graphs by extending the graph operations to Gaussian domain, bridging the interaction and aggregation across Gaussian groups.

• Experimental results illustrate that our method can generate efficient and generalizable Gaussian representations. Our model requires fewer Gaussians and achieves better rendering quality.

## 2 Related Works

### 2.1 Neural Implicit Representations

Early researches focus on capturing dense views to reconstruct scenes, while neural implicit representations have significantly advanced neural processing for 3D data and multi-view images, leading to high reconstruction and rendering quality [29, 35, 59, 40]. In particular, Neural Radiance Fields (NeRF) [30] has garnered considerable attention with a fully connected neural network to represent

complex 3D scenes. Subsequently, following works have emerged to address NeRF's limitations and enhance its performance. Some studies aim to solve the long-standing problem of novel view synthesis by improving the speed and efficiency of training and inference [15, 12, 31, 37, 26, 60, 10]. Other research focuses on modeling complex geometry and view-dependent effects to reconstruct dynamic scenes [7, 20, 36, 44, 51, 23, 48, 43, 21, 47]. Additionally, some studies [46, 42, 56, 52] have worked on reconstructing large urban scenes to avoid blurred renderings without fine details, which is a challenge for NeRF-based methods due to their limited model capacity. Furthermore, other works [33, 45, 50, 55] apply NeRF to novel view synthesis with sparse input views by incorporating additional training regularizations, such as depth, correspondence, and diffusion models.

## 2.2 3D Gaussian Splatting

More recently, 3D Gaussian Splatting (3DGS) [19] has drawn significant attention in the field of computer graphics, especially in the realm of novel view synthesis. Different from the expensive volume sampling strategy in NeRF, 3DGS utilizes a much more efficient rasterization-based splatting approach to render novel views from a set of 3D Gaussian primitives. However, 3DGS may suffer from artifacts, which occur during the splatting process. Thus, several works [57, 11, 17, 22] have been proposed to enhance the quality and realness of rendered novel views. Since 3DGS requires millions of parameters to represent a single scene, resulting in significant storage demands, some researches [27, 32, 13, 9, 18] focus on reducing the memory usage to ensure real-time rendering while maintaining the rendering quality. Other works [65, 54] are proposed to reduce the amount of required images to reconstruct scene. Besides, to extend the capability of 3D Gaussians from representing static scenes to 4D scenarios, several works [49, 8, 58, 53] have been proposed to incorporate faithful dynamics which are aligned with real-world physics.

## 2.3 Generalizable Novel View Synthesis

Optimization-based methods train a separate model for each individual scene and require dozens of images to synthesis high-quality novel views. In contrast, feed-forward models learn powerful priors from large-scale datasets, so that 3D reconstruction and view synthesis can be achieved via a single feed-forward inference. A pioneering work, pixelNeRF [61], proposes a feature-based method to employ the encoded features to render novel views. MVSNeRF [5] combines plane-swept cost volumes which is widely used in multi-view stereo with physically based volume rendering to further improve the quality of neural radiance field reconstruction. Subsequently, many researches [2, 3, 14, 16, 34, 38, 25] have developed novel view synthesis methods with generalization and decomposition abilities. Several generalizable 3D Gaussian Splatting methods have also been proposed to leverage sparse view images to reconstruct novel scenes. While Splatter Image [41] and GPS-Gaussian [63] regress pixel-aligned Gaussian parameters to reconstruct single objects or humans instead of complex scenes, pixelSplat [4] takes sparse views as input to predict Gaussian parameters by leveraging epipolar geometry and depth estimation. MVSplat [6] constructs a cost volume structure to directly predict depth from cross-view features, further improving the geometric quality.

However, these methods follow the paradigm of regressing pixel-aligned Gaussians and combine Gaussians from different views directly, resulting in an excessive number of Gaussians when the model processes multi-view inputs. In comparison, we construct Gaussian Graphs to model the relations of Gaussian groups. Furthermore, we introduce Gaussian Graph Network with specifically designed graph operations to ensure the interaction and aggregation across Gaussian groups. In this manner, we can obtain efficient and generalizable Gaussian representations from multi-view images.

## 3 Method

The overall framework is illustrated in Figure 2. After predicting positions and features of Gaussian groups from input views with extracted feature maps, we build a Gaussian Graph to model the relations. Then, we process the graph with Gaussian Graph Network via our designed graph operations on Gaussian domain for information exchange and aggregation across Gaussian groups. We leverage the fused Gaussians features to predict other Gaussian parameters, including opacity $\alpha$, covariance matrix $\Sigma$ and colors $c$.

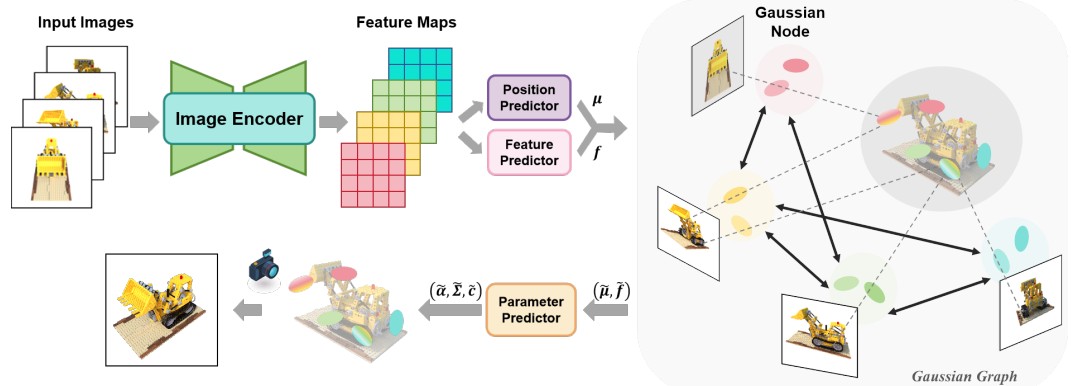

Figure 2: Overview of Gaussian Graph Network. Given multiple input images, we extract image features and predict the means and features of pixel-aligned Gaussians. Then, we construct a Gaussian Graph to model the relations between different Gaussian nodes. We introduce Gaussian Graph Network to process our Gaussian Graph. The parameter predictor generates Gaussians parameters from the output Gaussian features.

## 3.1 Building Gaussian Graphs

Given $N$ input images $\mathcal{I} = \{I_i\} \in \mathbb{R}^{N \times H \times W \times 3}$ and their corresponding camera parameters $\mathcal{C} = \{c_i\}$, we follow the instructions of pixelSplat [4] and MVSplat [6] to extract image features:

$$\mathcal{F} = \Phi_{image}(\mathcal{I}), \quad \mathcal{F} = \{F_i\} \in \mathbb{R}^{N \times \frac{H}{4} \times \frac{W}{4} \times C}, \tag{1}$$

where $\Phi_{image}$ is a 2D backbone. We predict the means and features of pixel-aligned Gaussians:

$$\mu_i = \psi_{unproj}\left(\Phi_{depth}(F_i), c_i\right) \in \mathbb{R}^{HW \times 3}, \quad f_i = \Phi_{feat}(F_i) \in \mathbb{R}^{HW \times D}, \tag{2}$$

where $\Phi_{depth}$ and $\Phi_{feat}$ stand for neural networks to predict depth maps and Gaussian features, and $\psi_{unproj}$ is the unprojection operation.

We build a Gaussian Graph $G$ with nodes $V = \{v_i\} = \{(\mu_i, f_i)\}_{1 \le i \le N}$. To model the relations between nodes, we define its adjacency matrix $A = [a_{ij}]_{N \times N}$ as:

$$a_{ij} = \begin{cases} 1, & i = j \\ \psi_{overlap}(v_i, v_j), & i \ne j \end{cases} \tag{3}$$

where $\psi_{overlap}(v_i, v_j)$ computes the overlap ratio between view $i$ and view $j$. To limit further computational complexity, we prune the graph by preserving edges with top $n$ weights and ignore other possible edges. The degree matrix $D = [d_{ij}]_{N \times N}$ satisfies $d_{ii} = \sum_j a_{ij}$. Thus, the scaled adjacency can be formulated as:

$$\tilde{A} = D^{-1}A \quad \text{or} \quad \tilde{A} = D^{-\frac{1}{2}}AD^{-\frac{1}{2}}. \tag{4}$$

## 3.2 Gaussian Graph Network

**Linear Layers.** Assuming that a conventional graph has $N$ nodes with features $\{g_i\} \in \mathbb{R}^{N \times C}$ and the scaled adjacency matrix $A = [\tilde{a}_{ij}]_{N \times N}$, the basic linear operation can be formulated as:

$$\hat{g}_i = \sum_{j=1}^{N} \tilde{a}_{ij} g_j W \in \mathbb{R}^D, \tag{5}$$

where $W \in \mathbb{R}^{C \times D}$ is a learnable weight. Different from the vector nodes $g_i$ in conventional graphs, each node of our Gaussian Graph contains a set of pixel-aligned Gaussians $v_i = \{\mu_i, f_i\}$. Therefore, we extend the scalar weight $a_{sk}$ of an edge to a matrix $E^{s \to k} = [e_{ij}^{s \to k}]_{HW \times HW}$, which depicts the detailed relations at Gaussian-level between $v_s$ and $v_k$. For the sake of simplicity, we define

$$e_{ij}^{s \to k} = \begin{cases} 1, & \psi_{proj}(\mu_{s,i}, c_s) = \psi_{proj}(\mu_{k,j}, c_s) \\ 0, & \psi_{proj}(\mu_{s,i}, c_s) \ne \psi_{proj}(\mu_{k,j}, c_s) \end{cases} \tag{6}$$

---

**Algorithm 1** Gaussian Graph Network

---

**Input:** Multi-view images $\mathcal{I} = \{I_i\}$, camera parameters $\mathcal{C} = \{c_i\}$, the number of graph layers $h$
**Output:** Gaussian parameters $(\mu, \Sigma, \alpha, c)$
    $\mathcal{F} \leftarrow \Phi_{image}(\mathcal{I}, \mathcal{C})$
    $\mu_i \leftarrow \psi_{unproj}\left(\Phi_{depth}(F_i), c_i\right), \quad f_i \leftarrow \Phi_{feat}(F_i) \quad i = 1, 2, \cdots, N$
    $\tilde{A} \leftarrow D^{-1}A$
    **for** $k \leftarrow 1$ to $h$ **do**
        $f_i \leftarrow \sigma\left(\sum \tilde{a}_{ij} E^{j \rightarrow i} f_j W\right) \quad i = 1, 2, \cdots, N$
    **end for**
    $(\mu, f) \leftarrow \bigcup_{s=1}^{l} \phi_{pooling}(G^s)$
    $R, S, \alpha, c \leftarrow \psi_R(f), \psi_S(f), \psi_\alpha(f), \psi_c(f)$
    $\Sigma \leftarrow RSS^\top R^\top$

---

where $\mu_{s,i}$ is the center of the $i$-th Gaussian in Gaussian group $v_s$, and $\psi_{proj}$ is the projection function which returns coordinates of the occupied pixel. In this manner, the linear operation on a Gaussian Graph can be defined as:

$$\hat{f}_i = \sum_{j=1}^{N} \tilde{a}_{ij} E^{j \rightarrow i} f_j W \in \mathbb{R}^D. \tag{7}$$

**Pooling layers.** If we have a series of connected Gaussian Graphs $\{G^s\}_{1 \leq s \leq l}$, where $G^s \subseteq G$, then we operate the specific pooling operation on each $G^s$ and combine them together:

$$\phi_{pooling}(G) = \bigcup_{s=1}^{l} \phi_{pooling}(G^s). \tag{8}$$

If $G^s$ only contains one node $v^s$, $\phi_{pooling}(G^s) = v^s$. Otherwise, we start with a random selected node $v_i^s = (\mu_i^s, f_i^s) \in G^s$. Since $G^s$ is a connected graph, we can randomly pick up its neighbor $v_j^s$ with corresponding camera parameters $c_j^s$. We define a function $\psi_{similarity}$:

$$\psi_{similarity}\left(v_{j,m}^s, v_i^s\right) = \begin{cases} \max_n \|\mu_{j,m}^s - \mu_{i,n}^s\|, & \max_n e_{mn}^{j \rightarrow i} = 1 \\ \infty, & \max_n e_{mn}^{j \rightarrow i} = 0 \end{cases} \tag{9}$$

where $v_{j,m}^s$ is the $m$-th Gaussian in node $v_j^s$. We define the merge function $\psi_{merge}$:

$$\psi_{merge}\left(v_{j,m}^s, v_i^s\right) = \begin{cases} \varnothing, & \psi_{similarity}\left(v_{j,m}^s, v_i^s\right) < \lambda \\ v_{j,m}^s, & \text{otherwise} \end{cases} \tag{10}$$

Then, we merge two nodes together to get a new node

$$v_{new}^s = \bigcup_{m=1}^{HW} \psi_{merge}\left(v_{j,m}^s, v_i^s\right) \cup v_i^s \tag{11}$$

In this manner, we aggregate a connected graph $G^s$ step by step to one node. Specifically, previous methods can be considered as a degraded Gaussian Graph without edges:

$$\phi_{pooling}(G) = \bigcup_{s=1}^{N} \phi_{pooling}(G^s) = \bigcup_{s=1}^{N} v^s. \tag{12}$$

where $\phi_{pooling}$ degenerates to simple combination of nodes.

**Parameter prediction.** After aggregating the Gaussian Graph $(\mu, f) = \psi_{pooling}(G)$, we predict the rotation matrix $R$, scale matrix $S$, opacity $\alpha$ and color $c$:

$$R = \phi_R(f), \quad S = \phi_S(f), \quad \alpha = \phi_\alpha(f), \quad c = \phi_c(f) \tag{13}$$

where $\phi_R$, $\phi_S$, $\phi_\alpha$ and $\phi_c$ are prediction heads. The covariance matrix can be formulated as:

$$\Sigma = RSS^\top R^\top. \tag{14}$$

Our Gaussian Graph Network is illustrated in Algorithm 1, where $\sigma$ stands for non-linear operations, such as ReLU and GeLU.

# 4 Experiments

We validate our proposed Gaussian Graph Network for novel view synthesis. In Section 4.1, we compare various methods on RealEstate10K [64] and ACID [24] under different input settings. In Section 4.2, we analyze the efficiency of our representations. In Section 4.3, we further validate the generalization of our Gaussian representations under cross dataset evaluation. In Section 4.4, we ablate our designed operations of Gaussian Graph Network.

Table 1: Quantitative comparison on RealEstate10K [64] benchmarks. We evaluate all models with 4, 8, 16 input views. [†] Models accept multi-view inputs, and only preserve Gaussians from two input views for rendering.

| Views | Methods | PSNR↑ | SSIM↑ | LPIPS↓ | Gaussians (K) | FPS↑ |
|---|---|---|---|---|---|---|
| 4 views | pixelSplat | 20.19 | 0.742 | 0.224 | 786 | 110 |
| | pixelSplat[†] | 20.84 | 0.765 | 0.2217 | 393 | 175 |
| | MVSplat | 20.86 | 0.763 | 0.217 | 262 | 197 |
| | MVSplat[†] | 21.48 | 0.768 | 0.213 | 131 | 218 |
| | Ours | 24.76 | 0.784 | 0.172 | 102 | 227 |
| 8 views | pixelSplat | 18.78 | 0.690 | 0.304 | 1572 | 64 |
| | pixelSplat[†] | 20.79 | 0.754 | 0.243 | 393 | 175 |
| | MVSplat | 19.69 | 0.768 | 0.238 | 524 | 133 |
| | MVSplat[†] | 21.39 | 0.766 | 0.215 | 131 | 218 |
| | Ours | 25.15 | 0.793 | 0.168 | 126 | 208 |
| 16 views | pixelSplat | 17.80 | 0.647 | 0.320 | 3175 | 37 |
| | pixelSplat[†] | 20.75 | 0.754 | 0.245 | 393 | 175 |
| | MVSplat | 19.18 | 0.753 | 0.250 | 1049 | 83 |
| | MVSplat[†] | 21.34 | 0.765 | 0.215 | 131 | 218 |
| | Ours | 26.18 | 0.825 | 0.154 | 150 | 190 |

Table 2: Quantitative comparison on ACID [24] benchmarks. We evaluate all models with 4, 8, 16 input views. [†] Models accept multi-view inputs, and only preserve Gaussians from two input views for rendering.

| Views | Methods | PSNR↑ | SSIM↑ | LPIPS↓ | Gaussians (K) | FPS↑ |
|---|---|---|---|---|---|---|
| 4 views | pixelSplat | 20.15 | 0.704 | 0.278 | 786 | 110 |
| | pixelSplat[†] | 23.12 | 0.742 | 0.219 | 393 | 175 |
| | MVSplat | 20.30 | 0.739 | 0.246 | 262 | 197 |
| | MVSplat[†] | 23.78 | 0.742 | 0.221 | 131 | 218 |
| | Ours | 26.46 | 0.785 | 0.175 | 102 | 227 |
| 8 views | pixelSplat | 18.84 | 0.692 | 0.304 | 1572 | 64 |
| | pixelSplat[†] | 23.07 | 0.738 | 0.232 | 393 | 175 |
| | MVSplat | 19.02 | 0.705 | 0.280 | 524 | 133 |
| | MVSplat[†] | 23.72 | 0.744 | 0.223 | 131 | 218 |
| | Ours | 26.94 | 0.793 | 0.170 | 126 | 208 |
| 16 views | pixelSplat | 17.32 | 0.665 | 0.313 | 3175 | 37 |
| | pixelSplat[†] | 23.04 | 0.694 | 0.279 | 393 | 175 |
| | MVSplat | 17.64 | 0.672 | 0.313 | 1049 | 83 |
| | MVSplat[†] | 23.70 | 0.709 | 0.278 | 131 | 218 |
| | Ours | 27.69 | 0.814 | 0.162 | 150 | 190 |

## 4.1 Multi-view Scene Reconstruction and Synthesis

**Datasets.** We conduct experiments on two large-scale datasets, including RealEstate10K [64] and ACID [24]. RealEstate10K dataset comprises video frames of real estate scenes, which are

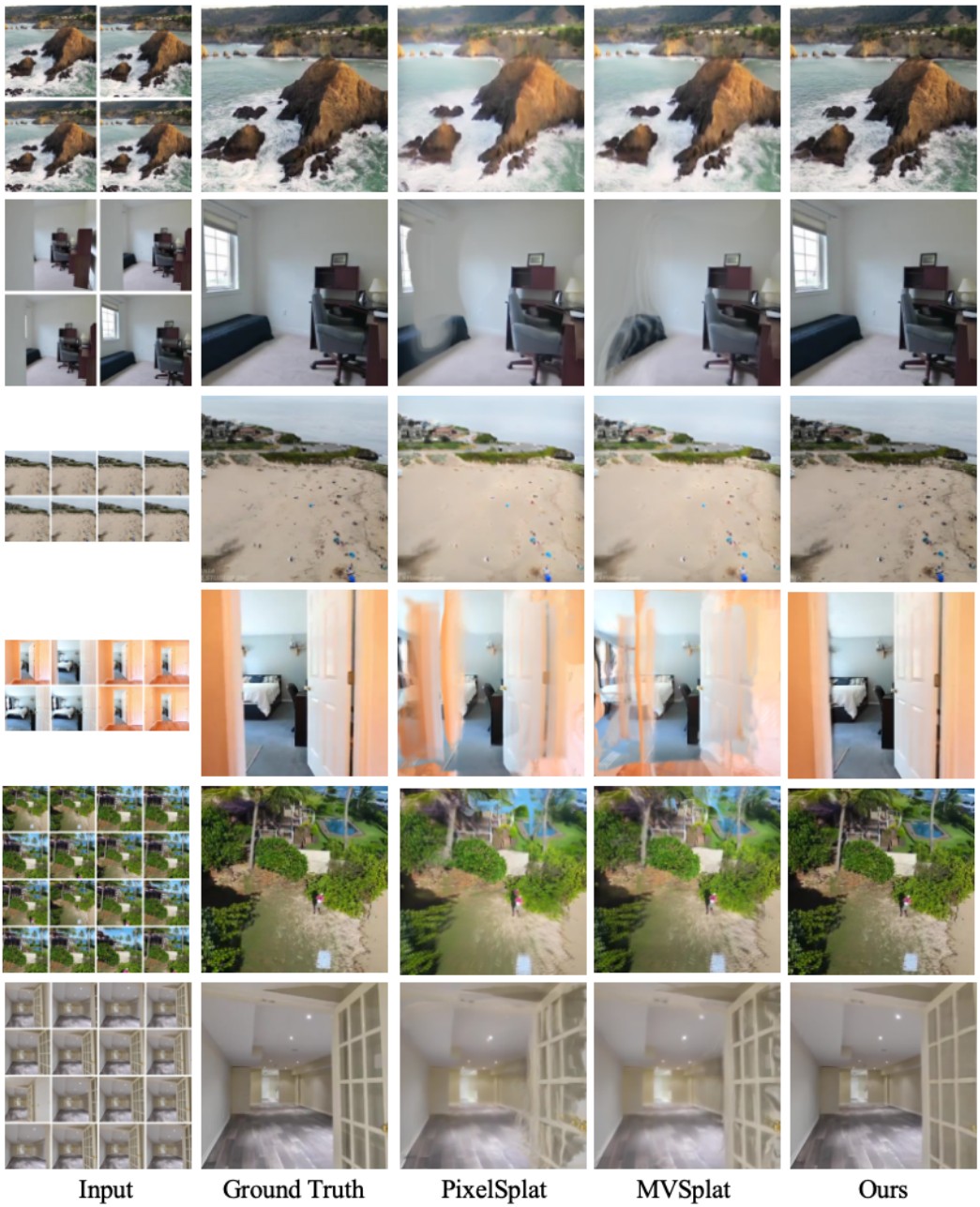

| Input | Ground Truth | PixelSplat | MVSplat | Ours |

Figure 3: Visualization results on RealEstate10K [64] and ACID [24] benchmarks. We evaluate all models with 4, 8, 16 views as input and subsequently test on three target novel views.

split into 67,477 scenes for training and 7,289 scenes for testing. ACID dataset consists of natural landscape scenes, with 11,075 training scenes and 1,972 testing scenes. For two-view inputs, our model is trained with two views as input, and subsequently tested on three target novel views for each scene. For multi-view inputs, we construct challenging subsets for RealEstate10K and ACID across a broader range of each scenario to evaluate model performance. We select 4, 8 and 16 views as reference views, and evaluate pixelSplat [4], MVSplat [6] and ours on the same target novel views.

**Implementation details.** Our model is trained with two input views for each scene on a single A6000 GPU, utilizing the Adam optimizer. Following the instruction of previous methods [4, 6], all experiments are conducted on $256 \times 256$ resolutions for fair comparison. The image backbone is initialized by the feature extractor and cost volume representations in MVSplat [6]. The number of

Table 3: Inference time comparison across different views. We train our model on 2 input views and report the inference time for 4 views, 8 views, and 16 views, respectively.

|  |  | 2 views | 4 views | 8 views | 16 views |
|---|---|---|---|---|---|
| pixelSplat [4] | 125.4 | 137.3 | 298.8 | 846.5 | 2938.9 |
| MVSplat [6] | 12.0 | 60.6 | 126.4 | 363.2 | 1239.8 |
| Ours | 12.5 | 75.6 | 148.1 | 388.8 | 1267.5 |

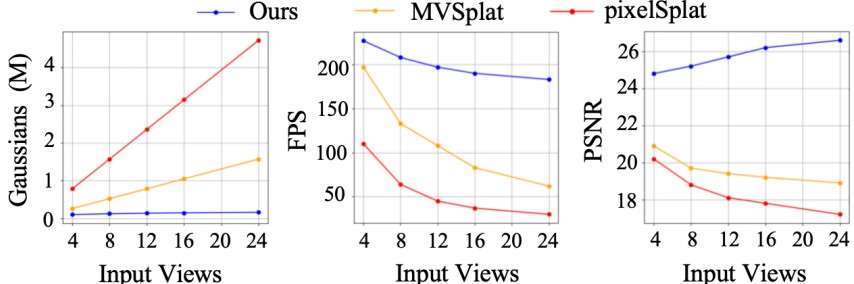

Figure 4: Efficiency analysis. We report the number of Gaussians (M), rendering frames per second (FPS) and reconstruction PSNR of pixelSplat [4], MVSplat [6] and our GGN.

graph layer is set to 2. The training loss is a linear combination of MSE and LPIPS [62] losses, with loss weights of 1 and 0.05.

**Results.** As shown in Table 1 and Table 2, our Gaussian Graph benefits from increasing input views, whereas both pixelSplat [4] and MVSplat [6] exhibit declines in performance. This distinction highlights the superior efficiency of our approach, which achieves more effective 3D representation with significantly fewer Gaussians compared to pixel-wise methodologies. For 4 view inputs, our method outperforms MVSplat [6] by about 4dB on PSNR with more than $2\times$ fewer Gaussians. For more input views, the number of Gaussians in previous methods increases linearly, while our GGN only requires a small increase. Considering that previous methods suffer from the redundancy of Gaussians, we adopt these models with multi-view inputs and preserve Gaussians from two input views for rendering. Under this circumstance, pixelSplat [4] and MVSplat [6] achieve better performance than before, because the redundancy of Gaussians is alleviated to some degree. However, the lack of interaction at Gaussian level limits the quality of previous methods. In contrast, our Gaussian Graph can significantly enhance performance by leveraging additional information from extra views. Visualization results in Figure 3 also indicate that pixelSplat [4] and MVSplat [6] tend to suffer from artifacts due to duplicated and unnecessary Gaussians in local areas, which increasingly affects image quality as more input views are added.

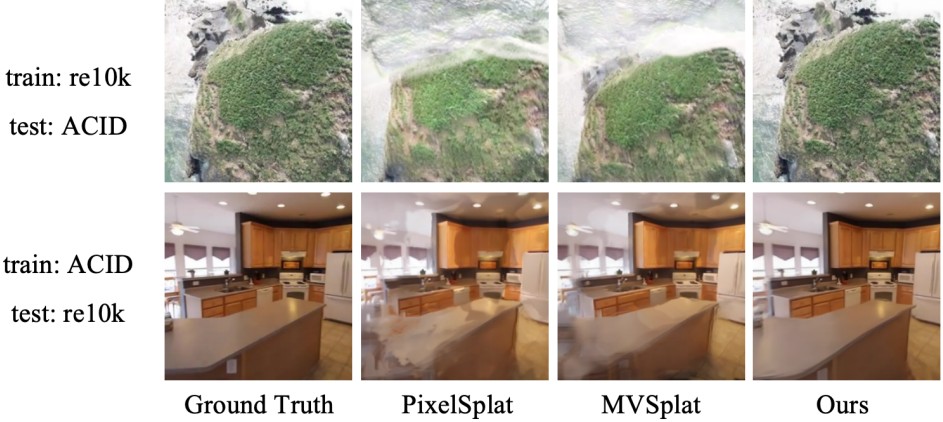

Figure 5: Visualization of model performance for cross-dataset generalization on RealEstate10K [64] and ACID [24] benchmarks.

Table 4: Cross-dataset performance and efficiency comparisons on RealEstate10K [64] and ACID [24] benchmarks. We assign eight views as reference and test on three target views for each scene.

| Train Data | Test Data | Method | PSNR↑ | SSIM↑ | LPIPS↓ | Gaussians (K) | FPS |
|------------|-----------|--------|-------|-------|--------|---------------|-----|
| ACID | re10k | pixelSplat | 18.65 | 0.715 | 0.276 | 1572 | 64 |
| | | MVSplat | 19.17 | 0.731 | 0.270 | 524 | 133 |
| | | Ours | 24.75 | 0.759 | 0.252 | 126 | 208 |
| re10k | ACID | pixelSplat | 19.75 | 0.724 | 0.264 | 1572 | 64 |
| | | MVSplat | 20.58 | 0.735 | 0.239 | 524 | 133 |
| | | Ours | 25.21 | 0.762 | 0.234 | 126 | 208 |

Table 5: Ablation study results of GGN on RealEstate10K [64] benchmarks. Each scene takes eight reference views and renders three novel views.

| Models | PSNR↑ | SSIM↑ | LPIPS↓ | Gaussians (K) |
|--------|-------|-------|--------|---------------|
| Gaussian Graph Network | 25.15 | 0.786 | 0.232 | 126 |
| w/o Gaussian Graph linear layer | 24.72 | 0.778 | 0.246 | 126 |
| w/o Gaussian Graph pooling layer | 20.25 | 0.725 | 0.272 | 524 |
| Vanilla | 19.74 | 0.721 | 0.279 | 524 |

## 4.2 Efficiency Analysis.

In addition to delivering superior rendering quality, our Gaussian Graph network also enhances efficiency in 3D representations. As illustrated in Figure 4, when using 24 input images, our model outperforms MVSplat by 8.6dB on PSNR with approximately one-tenth the number of 3D Gaussians and more than three times faster rendering speed. Additionally, we compare the average inference time of our model with pixel-wise methods in Table 3. Our GGN is able to efficiently remove duplicated Gaussians and enhance Gaussian-level interactions, which allows it to achieve superior reconstruction performance with comparable inference speed to MVSplat.

## 4.3 Cross Dataset Generalization.

To further demonstrate the generalization of Gaussian Graph Network, we conduct cross-dataset experiments. Specifically, all models are trained on RealEstate10K [64] or ACID [24] datasets, and are tested on the other dataset without any fine-tuning. Our method constructs the Gaussian Graph according to the relations of input views. As shown in Table 4, our GGN consistently outperforms pixelSplat [4] and MVSplat [6] on both benchmarks.

## 4.4 Ablations

To investigate the architecture design of our Gaussian Graph Network, we conduct ablation studies on RealEstate10K [64] benchmark. We first introduce a vanilla model without Gaussian Graph Network. Then, we simply adopt Gaussian Graph linear layer to model relations of Gaussian groups from multiple views. Furthermore, we simply introduce Gaussian Graph pooling layer to aggregate Gaussian groups to obtain efficient representations. Finally, we add the full Gaussian Graph Network model to both remove duplicated Gaussians and enhance Gaussian-level interactions.

**Gaussian Graph linear layer.** The Gaussian Graph linear layer serves as a pivotal feature fusion block, enabling Gaussian Graph nodes to learn from their neighbor nodes. The absence of linear layers leads to a performance drop of 0.43 dB on PSNR.

**Gaussian Graph pooling layer.** The Gaussian Graph pooling layer is important to avoid duplicate and unnecessary Gaussians, which is essential for preventing artifacts and floaters in reconstructions and speeding up the view rendering process. As shown in Table 5, the introduction of Gaussian Graph pooling layer improves the rendering quality by 4.9dB on PSNR and reduces the number of Gaussians to nearly one-fourth.

# 5 Conclusion and Discussion

In this paper, we propose Gaussian Graph to model the relations of Gaussians from multiple views. To process this graph, we introduce Gaussian Graph Network by extending the conventional graph operations to Gaussian representations. Our designed layers bridge the interaction and aggregation between Gaussian groups to obtain efficient and generalizable Gaussian representations. Experiments demonstrate that our method achieves better rendering quality with fewer Gaussians and higher FPS.

**Limitations and future works.** Although GGN produces compelling results and outperforms prior works, it has limitations. Because we predict pixel-aligned Gaussians for each view, the representations are sensitive to the resolution of input images. For high resolution inputs, *e.g.* $1024 \times 1024$, we generate over 1 million Gaussians for each view, which will significantly increase the inference and rendering time. GGN does not address generative modeling of unseen parts of the scene, where generative methods, such as diffusion models can be introduced to the framework for extensive generalization. Furthermore, GGN focuses on the color field, which does not fully capture the geometry structures of scenes. Thus, a few directions would be focused in future works.

**Acknowledgements.** This work was supported by the National Natural Science Foundation of China under Grant 62206147. We thank David Charatan for his help on experiments in pixelSplat.

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
