# OpenReview forum: "Gaussian Graph Network: Learning Efficient and Generalizable Gaussian Representations from Multi-view Images"
_NeurIPS.cc/2024/Conference — NeurIPS 2024 poster_

### Official Review · Reviewer_Xdv1 · 2024-06-17

**Soundness:** 2
**Presentation:** 2
**Contribution:** 2
**Rating:** 5
**Confidence:** 4

**Summary:**

This paper presents Gaussian Graphs to construct the relations of different Gaussian groups, introduces a Gaussian Graph Network to process Gaussian Graphs.

**Strengths:**

Experimental results are sufficient and convincing. This work is easy to follow.

**Weaknesses:**

1. As far as I can see, the main work of this paper includes the construction of Gaussian Graphs and Gaussian Graph Network. But they are very similar to the common GNN and workflow of GNN. The main innovation of this paper may be the combination of GNN and GS. However, many works that combine GNN and GS have been proposed, such as SAGS and Hyper-3DG. So I think this paper is of low innovation.
2. The authors provide only a few experimental results.

**Questions:**

1. As far as I can see, the main work of this paper includes the construction of Gaussian Graphs and Gaussian Graph Network. But they are very similar to the common GNN and workflow of GNN. What's the difference between this paper and other works that combine GNN and GS have been proposed, such as SAGS and Hyper-3DG?
2. Could this work be compared with NeRF-based generalizable architectures in the experiments?

**Limitations:**

I think this paper is of low innovation, as methods have combined GS and GNN. The construction of Gaussian Graphs and Gaussian Graph Network are very common.

---

> ### Author Rebuttal · Authors · 2024-08-07
>
> Thanks for your valuable comments. Below we address specific questions.
>
> **1. The difference between this paper and other works that combine GNN and GS**
>
> Thank you for pointing out these relevant works SAGS [1] and Hyper-3DG [2]. We would like to highlight our key differences with these works.
>
> One of the key differences is the construction of Gaussian graphs. SAGS sets each point as a node and constructs graphs for KNN points, while Hyper-3DG first obtains patch features by aggregating gaussian features within each patch, and then sets each patch as a node with its feature. However, SAGS is still an optimization-based method, where the Gaussian-level graph is less efficient due to the large number of Gaussian points, while the patch-level graph in Hyper-3DG fails to support message passing to exploit Gaussian-level relations.
>
> In our work, we propose a hierarchical method to construct Gaussian Graphs - we formulate each view as a node but also fully preserve the Guassian-level structures in each node. We fully exploit the important view-related information, i.e. (1) group of pixel-aligned Gaussians belonging to the same view and (2) projection relations between different views. This information enables us to capture accurate and sparse cross-view Gaussian-level relation in an efficient manner with the edge matrix $E^{i\rightarrow j}$.
>
> Based on our carefully-designed hierarchical structure, we specifically design linear layers for feature learning by extending the scalar weight of an graph edge to a matrix. Furthermore, we propose a new pooling strategy to fuse Gaussians based on their spatial positions to avoid the redundancy of Gaussians in pixelSplat and MVSplat, while SAGS and Hyper-3DG do not introduce pooling layers in their graph networks.
>
> [1] Ververas, E., Potamias, R. A., Song, J., Deng, J., & Zafeiriou, S. (2024). SAGS: Structure-Aware 3D Gaussian Splatting. arXiv preprint arXiv:2404.19149.
>
> [2] Di, D., Yang, J., Luo, C., Xue, Z., Chen, W., Yang, X., & Gao, Y. (2024). Hyper-3DG: Text-to-3D Gaussian Generation via Hypergraph. arXiv preprint arXiv:2403.09236.
>
> **2. Comparison with NeRF-based generalizable methods**
>
> We reported the comparison with NeRF-based methods on 2-view settings in Table 2 of our paper. We also conduct further experiments with different number of input views, as shown in the following table. Experimental results show that our method have better rendering quality and inference speed, which is benefit from our efficient and generalizable Gaussian representation learning.
>
>
> Table 1: Comparison of NeRF-based methods on RealEstate10K with different number of input views.
>
> |           | 4 view |                    | 8 view |                 | 16 view |                |
> |-----------|--------|--------------------|--------|-----------------|---------|----------------|
> |           | PSNR   | Inference Time     | PSNR   | Inference Time  | PSNR    | Inference Time |
> | pixelNeRF [3] | 21.82  | 5.38           | 21.84  | 5.77            | 21.85   | 6.48           |
> | MuRF [4]      | 24.30  | 0.36           | 24.78  | 0.55            | 25.56   | 1.79           |
> | Ours          | 24.76  | 0.14           | 25.15  | 0.44            | 26.18   | 1.40           |
>
> [3] Yu, A., Ye, V., Tancik, M., & Kanazawa, A. (2021). pixelnerf: Neural radiance fields from one or few images. In Proceedings of the IEEE/CVF conference on computer vision and pattern recognition (pp. 4578-4587).
>
> [4] Xu, H., Chen, A., Chen, Y., Sakaridis, C., Zhang, Y., Pollefeys, M., ... & Yu, F. (2024). Murf: Multi-baseline radiance fields. In Proceedings of the IEEE/CVF Conference on Computer Vision and Pattern Recognition (pp. 20041-20050).

---

> > ### Comment · Reviewer_Xdv1 · 2024-08-10
> >
> > Thank you for the new experimental results. You have addressed my concerns, and I have improved the final rating.

---

### Official Review · Reviewer_zafq · 2024-07-01

**Soundness:** 3
**Presentation:** 3
**Contribution:** 4
**Rating:** 7
**Confidence:** 3

**Summary:**

This paper introduces the Gaussian Graph Network (GGN), a novel approach for generalizable 3D Gaussian Splatting (3GDS) reconstruction. The authors identify a problem with previous generalizable 3DGS work: they regress pixel-aligned Gaussians and combine Gaussians from different views directly, resulting in an excessive number of Gaussians when the model processes multi-view inputs. To address this issue, they propose using a graph network that identifies the relationships between the Gaussians generated from different views and merges them if they are similar in positions and features. Thus, GGN can reduce the number of Gaussians for better efficiency and reconstruction quality. Experiments show that GGN outperforms baselines in both efficiency and quality and scales well when the input views increase.

**Strengths:**

1. The idea of merging redundant Gaussians in generalizable 3DGS is novel and practical. Identifying the relationships between Gaussians using a graph network to merge them is also a novel idea.
2. The results are satisfactory. Compared to the baselines, GGN achieves better reconstruction results with fewer Gaussians and scales well when the input views increase.
3. The paper is well-written and the experiments are complete.

**Weaknesses:**

1. The inference efficiency of GGN is not shown in the experiments. Since the number of Gaussians is quite large and increases with more input views, the efficiency of GGN is a concern. While the authors provide the results of rendering time and the number of merged Gaussians, these data do not reflect the inference efficiency of the model.

Overall, this paper introduces a novel and practical idea for generalizable 3DGS reconstruction. I think this paper pushes the boundaries of generalizable 3D reconstruction and should be accepted.

**Questions:**

1. How is the inference efficiency compared to the baselines?
2. How is the inference efficiency when the number of input views increases?

**Limitations:**

The limitations are discussed in the paper.

---

> ### Author Rebuttal · Authors · 2024-08-07
>
> Thanks for your valuable comments. We agree with you that the analysis of training and inference latency of our proposed method is important. We discuss this topic in our general rebuttal. We will add this discussion on training and inference latency in Section 4.2 (line 190) as you suggested.

---

### Official Review · Reviewer_LnYN · 2024-07-11

**Soundness:** 3
**Presentation:** 3
**Contribution:** 4
**Rating:** 6
**Confidence:** 5

**Summary:**

This paper proposes a Graph Neural Network architecture to model the relations between multi-view 3D Gaussians predicted by generalizable 3DGS methods. The proposed method works in particularly better for large number (e.g., 4, 8, 16 etc) of input images, compared with previous methods like pixelSplat and MVSplat that simply combine per-view pixel-aligned Gaussians. The proposed method effectively reduces the redundancy of 3D Gaussians when more input images are given. Experiments are conducted on standard RealEstate10K and ACID datasets and the proposed method performs significantly better than pervious methods for large number of input views.

**Strengths:**

This paper tackles a common problem in existing generalizable 3DGS models like pixelSplat and MVSplat, where they simply combine per-view 3D Gaussians. This leads to redundancy for more input images. The proposed solution could be applicable to different models and improve their performance.

The presentation is clear and the experiments are well-designed and extensive. The proposed method achieves strong results on standard benchmarks.

**Weaknesses:**

A few implementation details are not completely clear. For example, how are the input views selected for different number of input views? Are they sampled to be more densely within some predefined frame range or are they sampled to cover larger regions of the scene? This also leads to another question if the proposed method could work on large-scale scenes since the model could handle many frames now? For instance, if the inputs are 10-20 images capturing a room, could the full room be reconstructed reasonably? I am also wondering if the authors plan to release the source code, which I think would benefit the reproducibility for the research community.

For results of different number of input views in Table 1 and Figure 4, are they obtained with a single model, or the author trained a specific model for each specific number of input views?

For efficiency analysis, this paper reported the FPS metric. However, I assume the FPS is measured only for the rendering part right? How about the network inference time and memory consumption, especially for more input images? Since the proposed method is a feed-forward model, where the network forward time should also be considered for benchmarking efficiency.

**Questions:**

Please refer to the weaknesses.

**Limitations:**

Yes, the limitations are discussed.

---

> ### Author Rebuttal · Authors · 2024-08-07
>
> Thanks for your valuable comments. Below we address specific questions.
>
> **1. The selection of input views**
>
> As the number of input views increases, they are sampled from larger regions of the scene. Thank you for point out this question. We will clarify this setting in Section 4.1 (line 157) in our final version.
>
> **2. Large-scale scenes**
>
> We further evaluate our method on longer videos sampled from large-scale scenes in RealEstate10K. The visualization results are shown in Figure 1 in the one-page pdf. Our method can reconstruct large-scale outdoor scenes and render better novel views than previous methods, benefiting from multiple input views. We also validate our method on indoor full room scenes. With 16 images capturing a room, we reconstruct the full room reasonably, while pixelSplat and MVSplat still suffer from the redundancy of Gaussians, due to the fact that the large-scale scenes require more input views. We report the quantitative results of each scene in the following table for detailed comparison.
>
> Table 1: Quantitative comparison of different methods on large-scale scenes.
>
> | Sence ID | pixelSplat | MVSplat | Ours |
> |:--------:|:----------:|:-------:|:----:|
> |  Re10k: fea544b472e9abd1 |    12.66    |   12.88   |  18.88 |
> |  Re10k: 21e794f71e31becb |    15.04    |   14.76   |  20.62 |
> |  Re10k: de45926738229f67 |    17.13    |   20.21   |  25.07 |
> |  ACID: d4453ce709bd53e1 |     11.64    |   13.14   |  20.44 |
> |  ACID: ee09e048af8deba6 |    17.77    |   16.86   |  26.55 |
> |  ACID: 3fcb7b6b398b4064 |    22.75    |   24.55   |  32.03 |
>
> **3. Release of the source code**
>
> We will release our paper on arXiv and the source code on github if our paper is accepted.
>
> **4. Questions about the different number of input views in Table 1 and Figure 4**
>
> For results of different number of input views, they are obtained with a single model. In our opinion, it is convenient and flexible for users to use one model to process different number of input views. The results also demonstrate that our model has the capability to reconstruct scenes with different number of input views.
>
> **5. Efficiency analysis**
>
> We agree with you that the analysis of training and inference latency of our proposed method is important. We discuss this topic in our general rebuttal. We will add this discussion on training and inference latency in Section 4.2 (line 190) as you suggested.

---

> > ### Comment · Reviewer_LnYN · 2024-08-13
> >
> > I have read all the reviews and the rebuttal. First, I would like to thank the authors for taking time in preparing the rebuttal and providing additional clarifications in the discussion. Second, as the questions or concerns I had in my earlier review have all been answered or addressed, I keep my initial rating (Weak Accept). However, I would suggest to add the following missing information in the final version:
> >
> > - Details of view selection and evaluation for different number of input views
> > - Video results of more input views, which would make it easier to perceive the global consistency of the reconstruction. Although a few rendered images are provided in the rebuttal, it's less clear (compared to videos) how the model works for more input views.
> > - Time and memory consumption of different number of views, as shown in the global response.

---

> > > ### Author Response · Authors · 2024-08-14
> > > **Thanks for your valuable feedback**
> > >
> > > Thanks for your valuable suggestions. We will clarify the details of view selection and evaluation for different number of input views in Section 4.1. We will prepare a website page, which shows multiple video results, and we will release it with our final paper. We will also add the time and memory consumption in the global response to Section 4.2.

---

### Official Review · Reviewer_sL12 · 2024-07-15

**Soundness:** 3
**Presentation:** 4
**Contribution:** 3
**Rating:** 6
**Confidence:** 5

**Summary:**

The paper presents an incremental design built upon existing generalizable GS reconstruction frameworks (e.g., PixelSplat, MVSplat) to fusion overlapped pixel-aligned gaussian points from multiple images with a graph-based operator and pooling. This additional design presents a faster and better rendering effect than PixelSplat and MVSplat.

**Strengths:**

1. Overall, I agree that the proposed framework is a good choice for fusing pixel-aligned gaussian points from frameworks like PixelSplat or MVSplat. Grouping and fusing these points in 3D space with a graph-based strategy sounds reasonable and effective in solving the artifact issue caused by PixelSplat.

2. The visualization and comparison do present a better novel view synthesis effect with fewer gaussian points when compared with previous efforts like PixelSplat and MVSplat, which results in faster render speed compared with these two works.

**Weaknesses:**

***[Efficiency and Scalability]***

Although the paper presents a better rendering speed (a natural outcome of reducing gaussian points), it lacks a discussion regarding learning efficiency: the training latency of the proposed design. We know the point-aligned gaussian points are actually quite a large number, especially when increasing the number and resolution of images. However, classical graph-based methods are usually not efficient in handling a large number of points. Further, the limitation on training efficiency might further restrict the stability of the proposed design. Consequently, I hope the author can consider adding a discussion and benchmarking on a breakdown of training and inference latency of the additional graph-based network. It is fine if the current proposed design is not so efficient, but it is definitely worth a discussion.

On the other hand, the proposed graph network is fundamentally seeking a way to process and fuse points in 3D space. The point cloud processing community already has years of accumulation on handling this kind of data efficiently and effectively. One suggestion is to further enhance performance with the latest efforts in point cloud processing, such as Point Transformer V3.

**Questions:**

Already discussed in weaknesses regard to the training efficiency and scalability.

**Limitations:**

The authors also mentioned the limitation in model efficiency, especially when increasing the number and resolution of images. Yet this part is actually more important and worth some number to measure this issue. Detailed discussion in weaknesses.

---

> ### Author Rebuttal · Authors · 2024-08-07
>
> Thanks for your valuable comments. Below we address specific questions.
>
> **1. Training and inference latency**
>
> We agree with you that the analysis of training and inference latency of our proposed method is important.  We discuss this topic in our general rebuttal. We will add this discussion on training and inference latency in Section 4.2 (line 190) as you suggested.
>
> **2. Inspiration from point cloud processing community**
>
> Your suggestion does inspire us a lot. We further add a point cloud processing part from Point Transformer V3 to extract Gaussian features (Model A). As shown in Table 1, such design has improvement on performance. Due to the limitation of rebuttal time, we adopt a simple design, but this design still boost the performance. We argue that this is a promising direction, and we will consider this suggestion as one of our future works to further benefit from the progress in point cloud processing area.
>
> Table 1: Results of PSNR on RealEstate10K benchmark.
>
> |          |  Ours | Model A |
> |:--------:|:-----:|:-------:|
> |  4 views | 24.76 |  24.83  |
> |  8 views | 25.15 |  25.26  |
> | 16 views | 26.18 |  26.30  |

---

> > ### Comment · Reviewer_LnYN · 2024-08-11
> > **Efficiency comparison with Point Transformer**
> >
> > Thanks the authors for the response. The comparison with Point Transformer looks interesting (Table 2), I am wondering how the proposed GGN compares with Point Transformer in terms of inference time and memory consumption with respect to different views? It would be great if the authors could also report the time and memory comparisons in Table 2 as well, thanks.

---

> > > ### Author Response · Authors · 2024-08-11
> > >
> > > Thanks for your comments. We report the inference time (ms) as well as memory cost (GB) in the following table. Due to the limitation of rebuttal time, we only add a simple part to our model, which leads to a little increase on inference time and memory cost. We will consider how to fully utilize the efficient point cloud processing methods instead of such simple combination as our future topic to further promote our research. Thanks for your inspiration again.
> > > |      |Ours  |              |      |Model A|              |      |
> > > |------|------|--------------|------|-------|--------------|------|
> > > |Views |PSNR  |Inference Time|Memory|PSNR   |Inference Time|Memory|
> > > |4     | 24.76| 148.1        |4.8   |24.83  |157.3         |5.4   |
> > > |8     | 25.15| 388.8        |8.6   |25.26  |410.4         |10.1  |
> > > |16    | 26.18| 1267.5       |21.4  |26.30  |1334.6        |26.0  |

---

> ### Comment · Reviewer_sL12 · 2024-08-14
>
> Thanks for the rebuttal. My concerns are well addressed. Thus, I keep my original rate to accept this paper.
>
> BTW: I am also considering the possibility of combining point cloud processing with Gaussian splatting. We have spent many years exploring how to handle unstructured data effectively and efficiently, and it's encouraging to see that authors and reviewer LnYN are also interested in this potential.

---

> ### Comment · Reviewer_LnYN · 2024-08-14
>
> Thank you, Reviewer sL12, for your insightful comments. I completely agree with your points and would like to echo your thoughts. Indeed, I also feel that combining point cloud processing with Gaussian splatting has strong potential, and I am excited to see what our community will develop in this direction :)
>
> Best Regards,
>
> Reviewer LnYN

---

### Author Rebuttal · Authors · 2024-08-07

We thank all reviewers for their insightful feedback. We tried to address all the questions for each reviewer in the rebuttal session below.

Here we discuss about the **training and inference efficiency** as  mentioned by Reviewer sL12, LnYN and zafq. As the point-aligned gaussian points are quite a large number, if we set each gaussian point as a node in the graph, we do face the problem of inefficiency. In this paper, we formulate each view as a node but also fully preserve the Gaussian-level structures in each node. We exploit the important view-related information, i.e. (1) group of pixel-aligned Gaussians belonging to the same view and (2) projection relations between different views. This information enables us to capture accurate and sparse cross-view Gaussian-level relations in an efficient manner. Based on this, our specifically designed layers for the Gaussian Graph are operated on such Gaussian groups for information fusion.

We further conduct experiments to analysis the training and inference latency on RealEstate10K. All the experiments are conducted on a single NVIDIA A6000 GPU. Since the inference of pixelSplat with 16 input views cannot be done in parallel on a 48G A6000, we split them into batches to get Gaussians.

As shown in Table 1 and Table 2, our method uses less training and inference time compared with pixelSplat, benefiting from faster encoding and rendering speed. Due to the additional construction of the Gaussian Graph and application of Gaussian Graph Network, our method uses a little more training and inference time than MVSplat. We also report time consumption of each part of our methods in Table 3, where the Gaussian Graph Network dominates in time consumption and the rendering time increases slowly with more input views. As illustrated in Table 4, the inference time of three methods grows at a quadratic rate with the increase of image resolution due to pixel-aligned Gaussians.

For memory analysis, as shown in Table 5, our method has 80% off on training memory cost compared with pixelSplat. The comparison of inference memory cost between our method and pixelSplat draws to the similar conclusion. The structure of Gaussian Graph leads to a little increase of memory cost compared to MVSplat, which can be almost ignored as the number of views increases.

Table 1: Training time (h) of different methods.

|    Methods    | pixelSplat | MVSplat | Ours |
|:-------------:|:----------:|:-------:|:----:|
| Training time |     183    |    78   |  111 |


Table 2: Inference time (ms) of different methods with different number of input views.

|          | pixelSplat | MVSplat |   Ours  |
|:--------:|:----------:|:-------:|:-------:|
|  2 views |    137.3   |   60.6  |  75.6   |
|  4 views |    298.8   |  126.4  |  148.1  |
|  8 views |    846.5   |  363.2  |  388.8  |
| 16 views |   2938.9   |  1239.8 | 1267.5  |

Table 3: Inference time (ms) of different parts in our method.

|          | Image Encoder |   GGN   | Parameter Predictor | Rendering |
|:--------:|:-------------:|:-------:|:-------------------:|:---------:|
|  2 views |      8.37     |  28.21  |        34.51        |    4.47   |
|  4 views |     10.01     |  88.89  |        44.56        |    4.64   |
|  8 views |     14.09     |  311.24 |        58.75        |    4.75   |
| 16 views |     21.15     | 1165.65 |        75.88        |    4.84   |

Table 4: Inference time (ms) of different methods with different view resolutions.

|          | pixelSplat | MVSplat |   Ours  |
|:--------:|:----------:|:-------:|:-------:|
|  256$\times$256 |    137.3   |   60.6  |  75.6   |
|  512$\times$512 |    574.2   |   235.3 |  319.5  |
|  1024$\times$1024 |  2445.4  |   943.3 |  1329.9 |

Table 5: Training memory cost (GB) of different methods.

|        Methods       | pixelSplat | MVSplat | Ours |
|:--------------------:|:----------:|:-------:|:----:|
| Training Memory Cost |    37.5    |   5.9   |  7.7 |

Table 6: Inference memory cost (GB) of different methods with different number of input views.

|          | pixelSplat | MVSplat | Ours |
|:--------:|:----------:|:-------:|:----:|
|  2 views |     6.1    |   1.6   |  3.2 |
|  4 views |    11.8    |   3.2   |  4.8 |
|  8 views |    29.0    |   8.0   |  8.6 |
| 16 views |    73.7    |   20.0  | 21.4 |

Please kindly let us know if you have any follow-up questions or areas needing further clarification. Your insights are valuable to us, and we are ready to provide any additional information that could be helpful.

---

> ### Comment · Reviewer_LnYN · 2024-08-11
> **Time and memory cost of the image encoder**
>
> Thanks the authors for the detailed response. Could the authors share more details about how the image encoder accepts more input views since I guess the multi-view epipolar attention in both pixelSplat and MVSplat would cost a lot of time and memory when the number of views increase? Did the author perform standard multi-view epipolar attention (for example, pair-wise) or is there any modification made here? Thanks.

---

> > ### Author Response · Authors · 2024-08-11
> > **Correction of Table 3**
> >
> > Thanks for your comments. We follow the construction of the image encoder in MVSplat, which employs Swin Transformer and cost volume techniques to extract features. The cost volume part does cost a lot of time and memory when the number of views increase, as you mentioned. In Table 3 of our general rebuttal, we only reported the inference time of Swin Transformer as the image encoder and added the inference time taken by cost volume techniques to GGN, leading to uncorrect results. We sincerely apologize for this mistake. Thanks for your correction. We further conduct efficiency experiments and correct the inference time of each part more detailedly. Our GGN does not take much time as reported in Table 3, while the image encoder actually dominates in time consumption.
> >
> > |          | Image Encoder |             |        | GGN     |
> > |----------|---------------|-------------|--------|---------|
> > |          | Swin Trans.   | Cost Volume | All    |         |
> > | 2 views  | 8.37          | 14.31       | 22.68  | 7.52    |
> > | 4 views  | 10.01         | 57.38       | 67.39  | 22.07   |
> > | 8 views  | 14.09         | 245.94      | 260.03 | 58.62   |
> > | 16 views | 21.15         | 1028.98     | 1050.13| 131.97  |

---

> > > ### Comment · Reviewer_LnYN · 2024-08-12
> > >
> > > Thanks for your response. I am a bit confused. Initially I thought the "Image Encoder" contains everything before the GGN. Now it seems that the "Image Encoder" only refers to the feature extractor (Swin). However, from the new table, there is still one component missing: the U-Net used for depth estimation from the cost volume. Is it considered when measuring the inference time? In addition, it seems to me that the cost volume takes much more inference time than expected, since the cross-view attention in Swin is kind of similar to the cost volume computation (i.e., cross-view matching), but Swin does not take too much time. Could the authors further elaborate how to understand this behavior? Thanks.

---

> > > > ### Comment · Reviewer_LnYN · 2024-08-12
> > > >
> > > > Aha, it seems that the "Image Encoder" refers to everything before GGN right? The formatting of markdown makes it a bit confusing. If this is the case, could the authors show a breakdown of the inference time for each component in the Image Encoder? Thanks.

---

> ### Author Response · Authors · 2024-08-13
> **Inference time of each component in the image encoder**
>
> Thanks for your comments. We are sorry to confuse you due to the unclear markdown table. Below we address specific questions.
>
> **Q:** Is the U-Net used for depth estimation from the cost volume considered in the inference time?
>
> **A:** Yes. In the previous table, the time taken by the U-Net used for depth estimation is added to the inference time. Both the cost volume refinement and the depth refinement parts employ 2D U-Net. We report their inference time in our new table separately.
>
> **Q:** Why does the cost volume take much more inference time than expected?
>
> **A:** The cost volume construction contains two parts: (a) prepare projected data with respect to Eq. (2) in MVSplat, (b) compute the correlations. The part (a) dominates in the time consumption, since it computes projected features for each image pair and each depth candidates, leading to high inference time. The inference time cost by computing the correlations is not the same as the inference time of cost volume as we reported in the previous table. Actually, the inference time of computing correlations is much less than the inference time of part (a).
>
> **Q:** Does the image encoder refers to everything before GGN?
>
> **A:** It seems that the markdown table is not clear enough. $\Phi_{image}$ in Eq. (1) of GGN refers to multi-view feature extraction, cost volume construction and cost volume refinement. $\Phi_{depth}$ in Eq. (2) of GGN refers to depth estimation and depth refinement. We report the inference time of all these components in the following table. We will further clarify this part in our final version (line 113).
>
> |          | Multi-view feature extraction | Cost volume construction |        | Cost volume refinement | Depth estimation | Depth refinement|
> |----------|-------------------------------|--------------------------|--------|------------------------|------------------|----------------|
> |          |                               | prepare projected data   | compute the correlations |         | | |
> | 2 views  | 8.37          | 6.77       | 0.09  | 6.56   | 0.27 | 7.99 |
> | 4 views  | 10.01         | 50.11      | 0.19  | 6.89   | 0.27 | 8.08 |
> | 8 views  | 14.09         | 237.40     | 0.37  | 7.11   | 0.28 | 8.28 |
> | 16 views | 21.15         | 1022.59    | 0.96  | 7.37   | 0.29 | 8.56 |

---

> > ### Comment · Reviewer_LnYN · 2024-08-13
> >
> > Thanks the authors for the feedback. Everything is clear to me now and it's great to identify the bottleneck in the full pipeline.

---

### Comment · Area_Chair_hVD8 · 2024-08-10

Hi reviewers,

Thank you for your hard work in reviewing the paper!
Please check out the authors' responses and ask any questions you have to help clarify things by Aug 13.

--AC

---

### Decision · Program_Chairs · 2024-09-25

**Decision:**

Accept (poster)

**Comment:**

This paper is a clear paper. It receives 1x accept, 1x weak accept and 1x borderline accept. The reviewers agreed that the paper proposes a novel and practical idea of merging redundant Gaussians in generalizable 3DGS. The proposed framework is a good choice for fusing pixel-aligned gaussian points from frameworks like PixelSplat or MVSplat. Grouping and fusing these points in 3D space with a graph-based strategy sounds reasonable and effective in solving the artifact issue caused by PixelSplat. The experimental results are sufficient and convincing. This work is easy to follow. The weaknesses are mostly clarifications on the efficiency and scalability, and experimental settings are fully addressed by the authors during rebuttal and the discussion phase. The meta-reviewer follows the suggestions of the reviewers to accept the paper.